# Unveiling Prostate-Specific Membrane Antigen’s Potential in Breast Cancer Management

**DOI:** 10.3390/cancers17030456

**Published:** 2025-01-28

**Authors:** Lucia Motta, Marialuisa Puglisi, Giuliana Pavone, Gianmarco Motta, Federica Martorana, Michelangelo Bambaci, Demetrio Aricò, Paolo Vigneri

**Affiliations:** 1Department of Clinical and Experimental Medicine, University of Catania, 95123 Catania, Italy; lucia.motta@humanitascatania.it (L.M.); gianmarco.motta@humanitascatania.it (G.M.); federica.martorana@unict.it (F.M.); paolo.vigneri@unict.it (P.V.); 2Medical Oncology Unit, Humanitas Istituto Clinico Catanese, Misterbianco, 95045 Catania, Italy; 3School of Specialization in Medical Oncology, Department of Human Pathology “G. Barresi”, University of Messina, 98125 Messina, Italy; marialuisapuglisi94@gmail.com; 4Nuclear Medicine Unit, Humanitas Istituto Clinico Catanese, Misterbianco, 95045 Catania, Italy; michelangelo.bambaci@humanitascatania.it (M.B.); demetrio.arico@humanitascatania.it (D.A.)

**Keywords:** PSMA, breast cancer, PET/CT, radioligand therapy, Luthetium-177

## Abstract

Breast cancer (BC) remains a major global health challenge and finding new ways to improve its detection and treatment is essential. This study explores the potential of a protein called prostate-specific membrane antigen (PSMA), which is widely used in prostate cancer (PCa) imaging and therapy, for use in BC. While PSMA was initially believed to be exclusive to PCa, researchers have found that PSMA is also present in the blood vessels supplying BC, particularly in more aggressive types such as triple-negative breast cancer (TNBC). This study reviews existing research and clinical trials investigating how PSMA-targeting techniques, such as advanced imaging and radiotherapy, could enhance BC diagnosis and treatment. Early findings suggest that PSMA-based imaging may help detect BC more accurately, and PSMA-targeted therapies could offer a more focused approach to treating the disease while minimizing harm to healthy tissues. However, further studies are needed to confirm these benefits and determine how best to integrate PSMA into BC care. Ongoing research and clinical trials will be crucial in understanding whether PSMA can become a valuable tool in improving outcomes for BC patients.

## 1. Introduction

In recent years, theragnostic approaches have emerged as promising strategies for the diagnosis and treatment of various aggressive neoplasms. Notably, the prostate-specific membrane antigen (PSMA)-based diagnostic and therapeutic modalities have significantly enhanced the management of prostate cancer (PCa) patients [1]. PSMA is a type II trans-membrane glycoprotein (Figure 1) encoded by the FOLH1 (folate hydrolase 1) gene. It was first identified as a protein predominantly expressed in prostate epithelial cells, and its expression is markedly elevated in most prostate cancer cells [2,3]. Interestingly, PSMA is also found in some normal tissues, such as the salivary glands, and in the neovascular endothelium of various solid tumors, including breast cancer, kidney cancer, glioblastoma, and hepatocarcinoma [4,5,6,7,8]. This widespread expression underscores its critical role in angiogenesis [9,10,11].

In nuclear medicine, with regard to cancer, the theranostic approach has been a foundational aspect of the specialty. In 1947, Seidlin et al. published the first treatment of thyroid cancer using radioactive iodine, a radionuclide that has also been widely utilized as a diagnostic tool for thyroid cancer [12]. Similarly to radioactive iodine, PSMA is valuable for both imaging and therapeutic procedures in the management of prostate cancer (PCa). For diagnostic purposes, PSMA ligands, such as gallium-68 (^68^Ga) or fluorine-18 (^18^F), enable positron emission tomography (PET) imaging, which has become a standard diagnostic tool for PCa patients. High PSMA expression on PET/CT imaging helps identify patients eligible for treatment with PSMA-based radiopharmaceuticals labeled with beta- and alpha-emitting therapeutic radionuclides, such as lutetium-177 (^177^Lu-PSMA) and actinium-225 (^225^Ac-PSMA), respectively [13].

Notably, radionuclide therapy using 177Lu-PSMA-617 was investigated in the VISION phase 3 trial, which demonstrated significant improvements in imaging-based progression-free survival (PFS) and overall survival (OS) for patients receiving 177Lu-PSMA-617 [14]. Breast cancer presents a significant challenge in medical oncology, especially triple-negative breast cancer (TNBC). This subtype primarily affects younger women and is distinguished by its aggressive nature and higher likelihood of metastasis compared to other BC subtypes [13,15]. The aggressiveness of this disease often requires broad-spectrum, multi-drug chemotherapy regimens, with or without immunotherapy, as the primary treatment approach depending on PD-L1 expression [16,17]. These combination therapies are characterized by significant systemic toxicity and a tendency to develop drug resistance and drug–drug interactions with several molecules [18,19]. To potentially enhance the immune response against tumor cells, these regimens may be supplemented with checkpoint inhibitors [20]. For patients who progress to later stages of treatment, advanced therapies such as antibody–drug conjugates, including Sacituzumab govitecan and Trastuzumab deruxtecan, may be employed to target and deliver cytotoxic agents directly to cancer cells, thus improving treatment specificity and efficacy [20,21].

PSMA shows higher specificity compared to conventional biomarkers such as FDG in PET imaging, as its expression is more selective in tumor-associated neovasculature, particularly in aggressive subtypes like TNBC [22]. This specificity reduces background uptake, leading to improved contrast and more accurate lesion detection.

Emerging research has highlighted the expression of PSMA in TNBC, which opens new avenues for both diagnostic and therapeutic strategies. The integration of PSMA-PET imaging could provide more precise detection and staging of the disease. Additionally, the exploration of radiolabeled PSMA ligands holds promise for the development of targeted radiotherapy, potentially improving outcomes by selectively delivering radiation to PSMA-expressing cancer cells and minimizing damage to surrounding healthy tissues.

Taken together, this evidence supports investigation of PSMA for the diagnosis and treatment of BC patients; Figure 2 illustrates the rationale behind the potential application of this technology in the management of BC.

The aim of this review is to analyze and discuss the potential role of PSMA-based diagnostics and therapies in patients with breast cancer. We evaluated preclinical studies, case reports, and ongoing clinical trials to provide a comprehensive overview of how PSMA targeting may enhance diagnostic accuracy and therapeutic outcomes in this patient population.

## 2. Preclinical Studies

Preclinical studies have investigated PSMA expression in endothelial cells across various cancer types, including breast cancer.

Kashoa et al. explored PSMA expression using immunohistochemical staining on both normal and cancerous breast tissues, covering primary and metastatic sites. Their findings revealed that PSMA is notably expressed in the neovasculature associated with breast cancer and its metastases [23]. Several years later, Zhou et al. employed data from the Cancer Genome Atlas (TCGA) to assess the genetic alterations and expression patterns of FOLH1, a gene linked to the overexpression of PSMA RNA transcripts and protein, in 998 breast cancer patients. Their analysis demonstrated that PSMA protein was predominantly expressed in basal-like breast cancer compared to other subtypes, which frequently exhibited FOLH1 amplification [24].

Heesch et al. investigated PSMA and its isoforms at mRNA and protein levels in a panel of 12 different TNBC cell lines. They found that the full-length PSMA transcript was present in nearly all TNBC cell lines. This study further examined whether TNBC cells could induce an angiogenic state in a human endothelial cell line (HUVEC). The co-culture of TNBC cells with HUVEC notably induced PSMA expression in the endothelial cells. PSMA expression was also assessed using a 3D tumor spheroid model, where higher PSMA levels were observed under hypoxic conditions. Experiments with [68Ga]Ga-PSMA uptake in TNBC cell lines indicate lower binding affinity, while subsequent testing of [177Lu]Lu77-PSMA for its apoptotic effect showed that some TNBC cell lines, particularly in co-cultures with HUVECs, exhibited higher apoptotic rates than untreated samples [9].

Building on these results, Heesch et al. extended their analysis of PSMA overexpression to the TNBC vasculature using female mice implanted with TNBC xeno-graft. The mice were administered either a single dose of [177Lu]Lu-PSMA-I&T or fractionated dose given every seven days for a total of four administrations. A control group received 0.9% NaCl. Tumor growth was monitored using [18F] FDG-PET imaging. This study revealed that both single and fractionated doses of [177Lu]Lu-PSMA significantly inhibited tumor growth and reduced tumor size. The treated groups, particularly those receiving a single dose, exhibited prolonged median survival compared to the control group. Ex vivo TUNEL staining confirmed a higher level of apoptotic cells in the treated groups. The predominant expression of PSMA on tumor vasculature emphasized [177Lu]Lu-PSMA-I&T’s capability to disrupt tumor-associated endothelial cells and reduce blood supply to the tumor, highlighting its potential therapeutic benefit [25].

## 3. Clinical Evidence

Advancements in nuclear medicine, including the wider availability of radiopharmaceuticals and innovative procedures, have expanded its clinical applications. Positron Emission Tomography/Computed Tomography (PET/CT) plays a vital role not only in disease staging but also in assessing therapeutic response, with 18F-fluoro-deoxyglucose (18F-FDG) being the most widely used radiotracer.

Breast cancer imaging with [18F]FDG PET/CT is possible due to the frequent overexpression of glucose transporters GLUT 1–3 [26]. However, [18F]FDG uptake varies across breast cancer subtypes, reflecting differences in glucose metabolism. TNBC and HER2-enriched breast cancers typically exhibit high [18F]FDG uptake, whereas luminal breast cancers, particularly the luminal A subtype, exhibit lower uptake [27].

Building on the expression of PSMA in tumors beyond PCa, we present the available clinical studies that evaluate the diagnostic and therapeutic applications of PSMA in BC (Table 1). Notably, clinical cases have shown concordance between ^68^Ga-PSMA and ^18^F-FDG imaging in metastatic BC lesions [28,29,30,31,32].

Medina-Ornelas S. et al., showed that 68Ga-PSMA PET/CT detects more positive cases in TNBC compared to luminal A breast cancer [30], a result attributed to higher PSMA expression in less differentiated, higher-grade tumors [33].

While FDG-PET remains the standard imaging method, growing but limited evidence supports the use of PSMA-PET in breast cancer, especially in advanced stages. PSMA-PET offers advantages, such as identifying additional metastatic sites and providing insights into molecular heterogeneity. In patients with distant metastases, [18F]PSMA-1007 detected more metastatic lesions than [18F]FDG. This difference is mainly attributed to higher accumulation of [18F]PSMA in distant metastases, likely due to the role of neoangiogenesis in metastatic development [34]. Satheke et al. reported on PSMA imaging in 19 BC patients finding significantly higher mean SUVs for distant metastases (6.9 ± 5.7) compared to primary tumors/local recurrences (2.5 ± 2.6, *p* = 0.04) and lymph node metastases (3.2 ± 1.8, *p* = 0.011) [35]. Additionally, [18F]PSMA-1007 demonstrated superior lesion detectability in areas with high physiological [18F]FDG uptake, such as the brain and cranium [13].

Addressing molecular heterogeneity is crucial, with diverse PSMA expression levels being exhibited. Hormone receptor status may change over time, underscoring the importance of biopsies for metastatic disease, as recommended by clinical guidelines. PSMA PET imaging can help elucidate this molecular variability [29].

PSMA expression in breast cancer lesions, as demonstrated by 68Ga-PSMA-PET imaging, and its absence in normal vascular endothelium make PSMA a promising target for antiangiogenic therapy [36]. PSMA-targeting therapeutic agents can selectively target and destroy tumor-associated blood vessels, delivering high local radiation doses to overcome tumor resistance while sparing normal tissues [29].

Following the FDA approval of 177Lu-PSMA-617 for metastatic castration-resistant prostate cancer (mCRPC), interest is growing in extending PSMA-targeted radioligand therapy (RLT) to other cancers. A 2021 review by Uijen et al. summarized PSMA-based RLT applications in solid tumors beyond prostate cancer, including breast cancer. However, evidence remains limited [10].

In a Phase I trial by Von Hoff et al., docetaxel-encapsulated nanoparticles (BIND-014), specifically targeting PSMA, were tested. A patient with breast cancer in the trial showed a partial response to the therapy [37]. Additionally, Yuri Tolkach et al. provided clinical evidence for PSMA-targeted RLT in breast cancer reporting on a TNBC patient treated with [177Lu]Lu-PSMA-RLT following significant tracer uptake on imaging. Unfortunately, the patient experienced disease progression after the second treatment cycle, leading to therapy discontinuation. [10,33]. Currently, to the best of our knowledge, no additional case reports on PSMA-based RLT in breast cancer have been identified.

**Table 1 cancers-17-00456-t001:** Clinical evidence on the diagnostic and therapeutic applications of PSMA in BC.

First Author	Year	Evidence Type	N. of Patients	BC Subtype	Stage of BC	Goal	Results
Sevastian Medina-Ornelas [30]	2020	Retrospective study	21	TNBC: 5 Luminal: 10 HER2+: 6	100% mBC	Diagnostic	The ^68^Ga-PSMA-11 PET/CT demonstrated lower detection rates compared to ^18^F-FDG PET/CT (sensitivity: 84% vs. 99%) across the general population, except in TNBC patients, where all FDG-positive lesions were also PSMA-positive.
Mike Sathekge [35]	2017	Prospective study	19	TNBC: 6 Luminal: 7 NA: 6	63% mBC 37% lBC	Diagnostic	It was demonstrated that ^68^Ga-PSMA-11 PET is more sensitive in detecting metastatic lesions, particularly in TNBC, due to the significant role of neoangiogenesis in this histotype.
Natalia Andryszak [13]	2024	Prospective study	10	TNBC	90% mBC 10% NED	Diagnostic	A greater number of metastases, with higher SUVs, were detected using ^68^Ga-PSMA-11 PET compared to ^18^F-FDG PET/CT. Notably, 10 small brain metastases (4–7 mm) were identified exclusively with Gallium-based imaging.
Daniel D. Von Hoff [37]	2016	Phase I	58 total 1 BC	NA	100% mBC	Therapeutic	The study evaluated the response rate to BIND-014, a PSMA-targeted docetaxel nanoparticle formulation. A BC patient treated with this agent achieved a partial response (PR) after 21 cycles of therapy.
Yuri Tolkach [33]	2018	Case Report	1	TNBC	100% mBC	Diagnostic and Therapeutic	The first clinical case demonstrated PSMA receptor status via ^68^Ga-PSMA-PET/CT, followed by treatment with ^177^Lu-PSMA. Despite disease progression after two cycles, the patient exhibited good tolerance to the therapy.

Abbreviations: BC: breast cancer; CT: computed tomography; DRs: detection rate; FDG: Fluorodeoxyglucose; HER2: Human Epidermal Growth Factor Receptor 2; mBC: metastatic breast cancer; lBC: localized breast cancer; NA: not available; NED: non-evidence of metastasis; PET: Positron Emission Tomography; PR: partial response; PSMA: prostate-specific membrane antigen; SUV: standardized uptake value; TNBC: triple-negative breast cancer.

## 4. Future Directions

Given the promising potential of PSMA as both a diagnostic marker and a therapeutic target in breast cancer, numerous studies are underway to determine its precise utility in this context. One notable clinical trial, NCT05867615-LUBASKET [38], is currently in progress to evaluate the therapeutic role of PSMA. This Phase 2, single-arm basket trial is enrolling patients with Gallium-68 or Fluorine-18 PSMA-positive PET/CT scans. Participants will undergo treatment with 177Lu-PSMA. The administered dose depends on toxicity risk: patients without risk factors will receive 7.4 GBq of 177Lu-PSMA, while those with at least one risk factor will receive 5.5 GBq. Treatment consists of four cycles, administered every eight weeks.

In addition to this trial, several other clinical studies are exploring the diagnostic applications of PSMA in breast cancer (Table 2).

An ongoing study (NCT06586047) is assessing PSMA expression in metastatic TNBC patients using 18F-DCFPyL-PET/CT. The study will compare its ability to detect lesions with that of FDG PET/CT. It seeks to identify PSMA-avid lesions using 18F-DCFPyL, determine their overlap with FDG-avid lesions, and analyze standardized uptake values (SUVs) from both imaging methods. The goal is to explore the relationship between tumor aggressiveness (SUV on FDG PET/CT) and tumor angiogenesis (SUV on 18F-DCFPyL-PET/CT).

Another more comprehensive study (NCT04573231) is enrolling patients with HER2-negative, AR-positive metastatic BC. Starting from the hypothesis that PSMA expression correlates with resistance to anti-androgen therapies, the primary objective of this study is to evaluate PSMA expression using 18F-DCFPyL PSMA-based PET/CT. The secondary objective is to assess PSMA levels in circulating tumor cells (CTCs), and analyze diagnostic metastatic tissue. A notable aspect of the study involves measuring PSMA expression in CTCs after two weeks of treatment with Bicalutamide and Ribociclib. The study includes an additional secondary objective: to assess the correlation between PSMA expression and the clinical benefits of Bicalutamide and Ribociclib in patients enrolled in NCT030901. A single-center, prospective Phase II imaging trial (NCT06059469) is using PSMA-PET/CT to investigate PSMA expression in patients with progressive metastatic TNBC. The study evaluates the relationship between FDG and PSMA radiotracer uptake and explores the potential of Lu-177-labeled PSMA for molecular radionuclide therapy in refractory metastatic TNBC.

Additionally, a phase 1 feasibility trial (NCT04750473) is employing Fluciclovine and PSMA as radiotracers in PET/CT imaging, aiming to detect and stage lobular variant breast cancer and evaluate concordance and discordance rates between the two imaging methods.

Lastly, a pilot study (NCT05622227) is evaluating the diagnostic efficacy of 68Ga-P16-093, an innovative PSMA-targeting radiopharmaceutical. The study directly compares its performance with 18F-FDG in the same cohort of breast cancer patients. The objective is to determine the potential of 68Ga-P16-093 as an improved imaging agent for breast cancer detection by comparing the SUVmax of different tumor lesions derived from 68Ga-P16-093 and 18F-FDG.

## 5. Conclusions

To date, most research on PSMA expression has focused primarily on PCa, as the PSMA protein was initially thought to be prostate-specific [24]. However, in recent years, several studies have revealed that PSMA is also expressed in other malignancies, including BC, particularly within the tumor endothelium of newly formed vasculature [39]. Building on this premise, our review critically evaluates advancements in scientific research on PSMA-based diagnostic tools and therapies in BC, with a particular emphasis on TNBC. This BC subtype is unresponsive to conventional hormonal and HER2-targeted therapies and is characterized by an aggressive clinical behavior and poor prognosis [40]. These factors highlight the urgent need for novel biomarkers and therapeutic targets, such as PSMA, to improve outcomes for this challenging-to-treat patient population.

PSMA presents several advantages over conventional biomarkers, including higher specificity in tumor-associated neovasculature, particularly in aggressive subtypes like TNBC. This specificity leads to improved lesion detection and reduced background uptake during imaging. Furthermore, PSMA’s dual capacity as both a diagnostic and therapeutic target enables theranostic approaches that combine precise imaging with targeted radionuclide therapy, significantly enhancing personalized treatment strategies.

However, several challenges remain in translating PSMA-based approaches into routine BC management. PSMA-PET imaging has demonstrated lower sensitivity compared to standard techniques like FDG-PET, with reported low SUVmax values (2.5 to 6.9), which may limit its diagnostic utility in BC [10]. Additionally, PSMA-targeted RLT faces hurdles such as variability in PSMA expression among BC subtypes, potentially reducing ligand uptake and treatment efficacy [41]. Other significant barriers include logistical issues related to radioactive waste disposal, differing regulatory frameworks across countries, and elevated costs of theranostic approaches [11,42].

Despite these limitations, PSMA-based strategies remain promising and warrant further exploration. Future studies should prioritize the standardization of imaging protocols, including the development of consistent criteria for PSMA imaging, such as SUVmax thresholds, to improve the reproducibility and comparability of results across studies.

Another crucial area of focus is conducting larger-scale clinical trials to compare PSMA-based imaging and therapies against current standards like FDG-PET clarifying their diagnostic superiority and clinical relevance. Further investigation into the variability of PSMA expression across BC subtypes, including Luminal and HER2-enriched, is also essential to identify patient populations most likely to benefit from PSMA-targeted approaches. Notably, PSMA-based techniques appear to have lower sensitivity in lobular BC, underscoring the need for tailored applications [40].

An alternative application of PSMA involves the use of antibody drug conjugates (ADCs), which have already shown promise in other malignancies, such as PCa [43], and glioma [44]. Exploring the efficacy of combining PSMA-ADCs with existing therapeutic regimens could open new avenues for BC treatment.

Finally, health economics considerations must be addressed. Balancing the cost-effectiveness of PSMA-based theranostics with clinical accessibility is critical, particularly for a prevalent malignancy like BC. Developing strategies that align innovation with broader clinical utility will be essential for integrating these approaches into routine practice.

In conclusion, while PSMA-based diagnostics and therapies hold significant promise in BC, particularly TNBC, substantial challenges remain. The need for larger, well-designed trials, standardized imaging protocols, and optimized RLT regimens underscores the importance of continued research. Future investigations and upcoming trials will be essential to clarify the role of PSMA as a reliable diagnostic and therapeutic target in BC, ultimately improving personalized treatment strategies and patient outcomes.

## Figures and Tables

**Figure 1 cancers-17-00456-f001:**
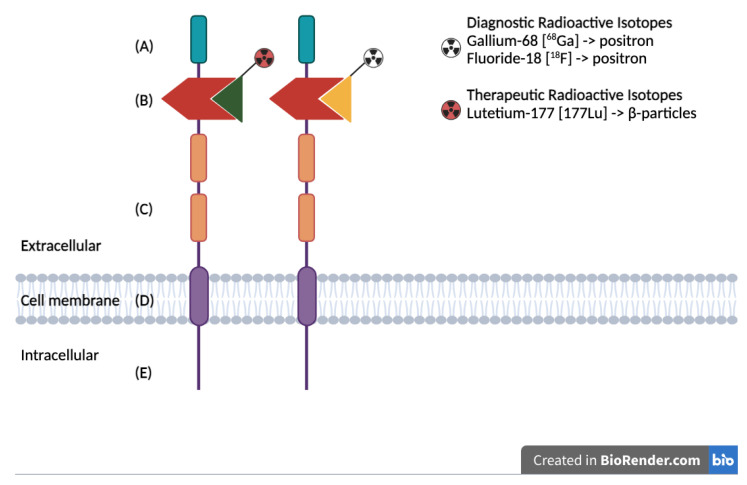
Schematic representation of the PSMA protein structure, highlighting the extracellular binding domain targeted for both diagnostic and therapeutic applications: (**A**) First extracellular domain of unknown function. (**B**) Catalytic domain with binding site for radiolabeled ligands. (**C**) Two domains of unknown function, with proline-rich and glycine-rich regions as linkers. (**D**) Hydrophobic domain. (**E**) Intracellular domain.

**Figure 2 cancers-17-00456-f002:**
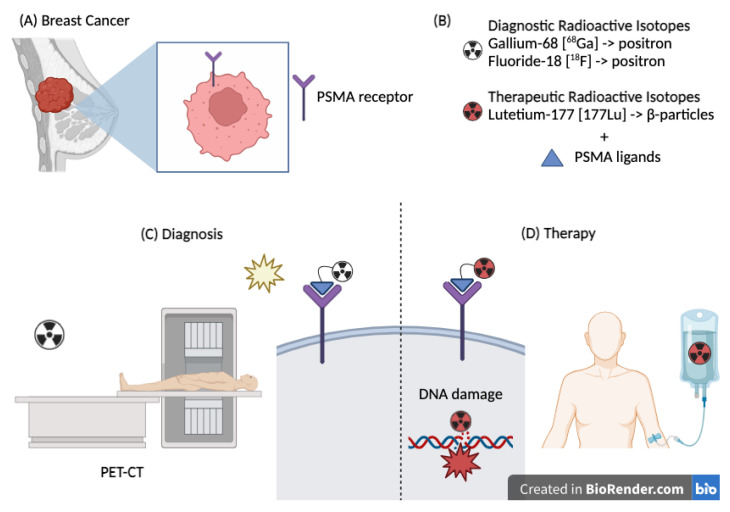
(**A**) Breast cancer expressing the PSMA receptor. (**B**) Radioisotopes used for diagnostic and therapeutic purposes. (**C**) Therapeutic use of PSMA–Lutetium causing direct damage to the DNA double helix. (**D**) Diagnostic use of PSMA (Gallium or Fluorine) through a PET scan.

**Table 2 cancers-17-00456-t002:** Ongoing trial regarding diagnostic applications of PSMA in BC.

Name	CT.gov ID	Phase	Goals
BC PSMA PET	NCT06586047	NA	Assess PSMA expression in TNBC patients using 18F-DCFPyL and correlate SUVs with prognosis
Evaluation of PSMA in HER2-AR+ Metastatic BC	NCT04573231	2	Assess PSMA expression in HER2-/AR+ mBC using 18F-DCFPyL, in CTCs, in the diagnostic tissue, and in CTCs after Bicalutamide and Ribociclib treatments
PSMA-PET/CT to Assess the Expression of Specific Membrane Antigen (PSMA) in Patients With Progressive TNBC (PRISMA)	NCT06059469	2	Correlate FDG and PSMA-labeled radiotracer uptake in progressive TNBC
Fluciclovine and PSMA PET/CT for the Classification and Improved Staging of Invasive Lobular BC	NCT04750473	1	Use Fluciclovine and PSMA as radiotracers in PET/CT imaging, aims to detect and stage lobular variant BC
68Ga PET/CT Imaging in Breast Cancer Patients	NCT05622227	1/2	Compare the tumor detected by 68Ga-P16-093 and 18F-FDG as radiotracer in BC

Abbreviations: AR: androgen receptors; BC: breast cancer; CT: computed tomography; CTCs: circulating tumor cells; FDG: Fluorodeoxyglucose; HER2: Human Epidermal Growth Factor Receptor 2; mBC: metastatic breast cancer; NA: not applicable; PET: Positron Emission Tomography; PSMA: prostate-specific membrane antigen; SUV: standardized uptake value; TNBC: triple-negative breast cancer; 18F-DCFPyL: Piflufolastat F 18; 68Ga: Gallium 68.

## Data Availability

No new data were created or analyzed in this study.

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
