# Peer review of "Unveiling Prostate-Specific Membrane Antigen’s Potential in Breast Cancer Management"

_cancers, 2025, doi:10.3390/cancers17030456_

Round 1
Reviewer 1 Report
Comments and Suggestions for Authors
1. What does the description of “PSMA is conjugated to various radionuclides…” mean? Did the authors want to say “PSMA ligands” instead?
2. The authors mentioned that “The absence of specific biomarkers for targeted therapies in this population….”, but there are already many reviews published focusing on the biomarkers for the prognosis and response prediction of TNBC, e.g. 10.5306/wjco.v6.i6.252; 10.1016/j.critrevonc.2019.102855; 10.3390/ijms21134579; 10.1038/s41419-019-2043-x; 10.3390/diagnostics13111949; 10.3390/ijms25052559.
3. What is the main difference of this manuscript comparing with the previously published reviews on the similar topic?
4. Are there any potential advantages of PSMA over other biomarkers that should be particularly emphasized?
Author Response
Comments 1: What does the description of “PSMA is conjugated to various radionuclides…” mean? Did the authors want to say “PSMA ligands” instead?
Response 1: We agree with the reviewer that PSMA ligand is a better definition. We modified the text accordingly (page number 2, paragraph 1, line 50)
Comments 2: The authors mentioned that “The absence of specific biomarkers for targeted therapies in this population….”, but there are already many reviews published focusing on the biomarkers for the prognosis and response prediction of TNBC, e.g. 10.5306/wjco.v6.i6.252; 10.1016/j.critrevonc.2019.102855; 10.3390/ijms21134579; 10.1038/s41419-019-2043-x; 10.3390/diagnostics13111949; 10.3390/ijms25052559.
Response 2: We thank the reviewer for this insightful observation. We rephrased the sentence to clarify that TNBC may present recently identified biomarkers, such as PD-L1. This revision has been incorporated in the introduction (page 2, paragraph 1, lines 62–64).
Comments 3: What is the main difference of this manuscript comparing with the previously published reviews on the similar topic?
Response 3: In our opinion, this review provides a timely, updated, and well-organized synthesis of the current knowledge on PSMA in breast cancer. Unlike previous reviews, it focuses exclusively on breast cancer, particularly emphasizing recent advancements in both diagnostic and therapeutic strategies. We particularly focused on the triple-negative breast cancer, which is known for its aggressive behavior and limited targeted treatment options. PSMA is regarded as a potential strategy for both diagnosis and therapy in this context. Furthermore, the review contrasts PSMA-PET imaging with established techniques like FDG-PET, discussing their respective advantages and limitations in breast cancer imaging, an aspect often underrepresented in prior literature.
Comments 4: Are there any potential advantages of PSMA over other biomarkers that should be particularly emphasized?
Response 4: We thank the reviewer for giving us the opportunity to clarify this point. PSMA displays several advantages over other biomarkers. These advantages have been emphasized in the revised version of our manuscript potential advantages (page number 10, paragraph 5, line 261-266). PSMA shows greater specificity compared to conventional biomarkers such as FDG in PET imaging, as its expression is more selective in tumor-associated neovasculature, particularly in aggressive subtypes like triple-negative breast cancer (TNBC). This specificity reduces background uptake, leading to improved contrast and more accurate lesion detection. Furthermore, PSMA's dual role as both a diagnostic and therapeutic target allows for theranostic approaches, enabling simultaneous imaging and targeted radionuclide therapy. These features make PSMA a highly promising biomarker compared to more generalized markers currently used in breast cancer imaging and therapy.
Reviewer 2 Report
Comments and Suggestions for Authors
General comment:
This work revise the role of PSMA in breast cancer providing a clear, concise and exhaustive overview of this field.
Specific comments throughout the paper:
The table for the trials is welcomed and relevant, since it summarizes the most relevant features. A similar summary table could be provided for the clinical evidence section, thus favoring the readers’ understanding.
The conclusion section should have more discussion points.
Author Response
Comments 1: The table for the trials is welcomed and relevant, since it summarizes the most relevant features. A similar summary table could be provided for the clinical evidence section, thus favoring the readers’ understanding
Response 1: Thank you for this suggestion. To further enhance clarity and facilitate the reader's understanding, we have added a summary table in the clinical evidence section (page number 7, paragraph 3, line 191).
Comment 2: The conclusion section should have more discussion points.
Response 2: Thank you for your valuable suggestion. In response, we have significantly revised and expanded the conclusion section to address your concerns and incorporate additional discussion points (page number 10, paragraph 5, line 261-294). Specifically, we have elaborated on PSMA's dual role as a diagnostic and therapeutic target, emphasizing its specificity in tumor-associated neovasculature, particularly in aggressive subtypes like TNBC, and its potential for theranostic applications. The revised conclusions discuss the challenges associated with PSMA-based imaging and therapy, such as the lower sensitivity of PSMA-PET compared to FDG-PET, the variability of PSMA expression across breast cancer subtypes, and the logistical and economic barriers to implementation. We have added new points highlighting the importance of standardizing imaging protocols, conducting larger comparative clinical trials, and refining biomarker-based patient stratification to improve clinical applicability. Lastly, we emphasized the need for balancing innovation with cost-effectiveness to make PSMA-based approaches accessible in routine clinical practice. These revisions aim to provide a more comprehensive, forward-looking discussion that addresses the critical challenges and opportunities in PSMA-based breast cancer management. We hope these additions meet your expectations and enhance the manuscript's contribution to the field
Reviewer 3 Report
Comments and Suggestions for Authors
cancers-3405404
Unveiling PSMA's Potential in Breast Cancer Management
1. The manuscript summarized the potential of PSMA-based radiopharmaceuticals for BC diagnosis and treatment through preclinical data, case reports, and ongoing clinical trials. Overall, the manuscript was appropriately prepared. However, with limited information, it may not be sufficiently novel and contribute to the field. It is simply a summary of several preclinical studies, case reports, and ongoing clinical trials.
2. There are only four preclinical studies and few clinical studies. All clinical trials have no current outcome to evaluate. I think this is not an appropriate time to conduct such a review and conclude anything about the topic.
3. The information presented is limited and unsuitable for a top-tier journal like Cancers.
4. Introduction: Figure 1 was simply introduced without discussion.
Author Response
Comments 1: The manuscript summarized the potential of PSMA-based radiopharmaceuticals for BC diagnosis and treatment through preclinical data, case reports, and ongoing clinical trials. Overall, the manuscript was appropriately prepared. However, with limited information, it may not be sufficiently novel and contribute to the field. It is simply a summary of several preclinical studies, case reports, and ongoing clinical trials.
Response 1: We appreciate the feedback. Based on our literature research, this is the first review that systematically and comprehensively organizes all available evidence on PSMA-based radiopharmaceuticals while focusing exclusively on breast cancer, a unique aspect compared to prior reviews. Nonetheless, we have further expanded our discussion section (page number 10, paragraph 5, line 261-294) to provide a more exhaustive and critical analysis, emphasizing both the current limitations and the potential advantages of PSMA-based strategies compared to standard diagnostic and therapeutic tools. We have also added new insights into the variability of PSMA expression across breast cancer subtypes, and the need for standardized imaging protocols.
Comments 2: There are only four preclinical studies and few clinical studies. All clinical trials have no current outcome to evaluate. I think this is not an appropriate time to conduct such a review and conclude anything about the topic
Response 2: Thank you for your insightful feedback. We acknowledge the limited number of preclinical and clinical studies available on this topic. However, research about PSMA in breast cancer is a rapidly evolving field with growing interest, as evidenced by the increasing number of ongoing clinical trials. We aim to provide a comprehensive synthesis of the current evidence, despite its preliminary nature. While definitive conclusions cannot yet be drawn, we believe it is valuable to consolidate the available findings at this stage to highlight the potential of PSMA as a promising biomarker for both diagnostic and therapeutic applications in breast cancer. We hope this review will serve as a useful reference point for guiding future studies and clinical trials in this emerging area.
Comments 3: The information presented is limited and unsuitable for a top-tier journal like Cancers
Response 3: Thank you for your candid feedback. While we acknowledge that the field of PSMA-based applications in breast cancer is still in its early stages, we believe this review provides significant value by organizing and synthesizing the available evidence in a structured and focused manner. This includes preclinical studies, clinical reports, and ongoing trials, which we have critically evaluated to identify potential advantages, limitations, and future directions for PSMA in breast cancer. We have enriched the discussion section (page number 10, paragraph 5, line 261-294) to address the current limitations and potential clinical implications of PSMA-targeted strategies. This includes a detailed analysis of its dual diagnostic and therapeutic potential, as well as considerations for health economics and biomarker variability. We have incorporated additional references and a new figure (page number 2, paragraph 1, line 39) to provide a visual and structural overview of PSMA applications, as suggested by another reviewer. While the field is still evolving, we believe our review serves as a valuable foundation for future research and highlights the importance of this emerging topic in breast cancer management. We hope these revisions address your concerns and demonstrate the manuscript’s contribution to the field.
Comments 4: Introduction: Figure 1 was simply introduced without discussion
Response 4: Thank you for pointing out this issue. We have revised the manuscript to provide a clearer explanation of Figure 2 (a new figure was added in response to another reviewer's suggestion, which is why we now refer to it as Figure 2). We have included a statement in the text (page 3, paragraph 1, line 85-87) to better integrate the figure into the discussion. We believe this clarification enhances the reader's understanding of the figure's relevance and its contribution to the manuscript.
Reviewer 4 Report
Comments and Suggestions for Authors
The authors provided an overview of the potential uses of PSMA-based radiopharmaceuticals for diagnosing and treating BC. Their review encompasses preclinical and clinical studies, along with ongoing trials, to explore PSMA-targeted approaches in BC management. Nevertheless, current data is not sufficient to conclusively establish whether PSMA-based imaging or therapeutic methods offer substantial advantages in BC.
The following are the suggestions/corrections to the authors before considered for publication.
1. While the introduction was well-presented, it is suggested that the authors incorporate a PSMA structure or image to enhance reader engagement.
2. Additionally, it is recommended to include recent publications, such as: 1) Int. J. Mol. Sci. 2024, 25(12), 6519n and 2) BMC Cancer, 2024 Oct 29;24:1328.
3. There is a discrepancy between the citation in line 156 (ref. 32) and the corresponding entry in the reference section. This inconsistency should be addressed.
4. The word 'Objective' should be corrected to 'objective'.
Author Response
Comments 1: While the introduction was well-presented, it is suggested that the authors incorporate a PSMA structure or image to enhance reader engagement.
Response 1: Thank you for your thoughtful suggestion. We have addressed this point by incorporating a figure (Figure 1, page number 2, paragraph 1, line 39) illustrating the structural representation of PSMA in the introduction section. This addition aims to enhance reader engagement and provide a clearer visual context for understanding the molecular features of PSMA. We hope this modification improves the clarity and visual appeal of the manuscript.
Comments 2: Additionally, it is recommended to include recent publications, such as: 1) Int. J. Mol. Sci. 2024, 25(12), 6519n and 2) BMC Cancer, 2024 Oct 29;24:1328.
Response 2: We agree that these recently published papers fit with our work and therefore we included them in our references’ list.
Comments 3: There is a discrepancy between the citation in line 156 (ref. 32) and the corresponding entry in the reference section. This inconsistency should be addressed.
Response 3: Thank you for pointing out this discrepancy. We corrected the typo.
Comments 4: The word 'Objective' should be corrected to 'objective'
Response 4: Thank you for your observation. We have corrected the capitalization of 'Objective' to 'objective' as suggested. The change has been applied throughout the manuscript where necessary.
Round 2
Reviewer 3 Report
Comments and Suggestions for Authors
None